# Invited perspectives: The ECMWF strategy 2021-2030 Challenges in the area of natural hazards

Florian Pappenberger[1], Florence Rabier[1], Fabio Venuti[1]

[1]European Centre for Medium-range Weather Forecasts, Reading/Bologna/Bonn, UK/Italy/Germany

*Correspondence to*: Florian Pappenberger (florian.pappenberger@ecmwf.int, @FPappenberger)

**Abstract.**

The European Centre for Medium-Range Weather Forecasts mission is to deliver high quality global medium-range numerical weather predictions and monitoring of the Earth system to its Member States. The modelling and forecasting of natural hazards are an important part of this mission. Challenges in this area include the integration of innovative observations into the Earth system, realistic representations of water, energy, and carbon cycles, coupling and initialisation of all Earth system components, adequate representation of uncertainties, supporting the development of user specific products to enable optimal decision making under uncertainties and advances in software engineering. The new ECMWF strategy identified 3 pillars to sustain its future development (ECMWF, 2021): Science and Technology (World leading weather and Earth system science, cutting-edge technology and computational science), Impact (High-quality products fit for purpose, efficient and easy access to products) and People (Inspiring and hiring the best experts). Progress in all these areas will need enhanced collaboration with Member States and partners across Europe and beyond.

## 1 Introduction

The forecasting of weather, hydrology, cryosphere, and oceans, as well as the representation of past and future climates, are all key elements in the understanding and prediction of natural hazards. Forecasts of the Earth system are important to prepare for humanitarian disasters resulting from, for example, tropical cyclones (Emerton et al. 2020, Magnusson, 2019), floods (Lavers et at 2020), extreme precipitation events (Lavers et al 2018) or heat waves (Napoli et al 2020). Representations of the past climate, known as reanalysis, are key to establish risk i.e., return periods (Harrigan et al (2020a), natural and artificial trends (Zsótér et al 2020) as well as understanding scientific challenges. Progress in these areas can only be achieved through an Earth system approach (Harrigan et al 2020b).

The vision is to produce cutting-edge science, world-leading weather predictions and monitoring of the Earth system with a particular focus on the medium and sub-seasonal forecast range. To meet this vision, there are several research challenges that need addressing. The required step changes in science and forecasting can be grouped into 3 pillars (ECMWF, 2021): Science and Technology (World leading weather and Earth system science, cutting-edge technology and computational

science), Impact (High-quality products fit for purpose, efficiently and easily accessed) and People (Inspiring and hiring the best experts).  ECMWF has been a collaborative organisation from the start and the size of the upcoming challenges make it even more important to collaborate effectively with its Member States.

## 2 Science and Technology

Earth system model predictions can be substantially improved if we maximise the use of current observations (Figure 1) e.g.,
using all available information from existing satellites over land, snow and sea ice in cloudy, rainy and clear conditions (Geer et al., 2018). Newly available observations from satellites and radar, will need to be efficiently integrated into the Earth system model. For example, the new EUMETSAT MTGs will provide high frequency atmospheric profiles as well as real-time lightning detection (lightnings can cause fires). The European Commission's Copernicus Sentinel 6 Michael Freilich (a radar altimeter satellite) will measure sea level change and river levels addressing a range of natural hazards such
as floods, storm surges and droughts.  In addition, the operational use of innovative new observing systems, e.g., derived through the Internet of Things (IoT, including observations from cars and mobile phones), provides the potential to significantly advance science and forecasts of the Earth system and natural hazards. Collaboration with the Member States in this area will be particularly useful, as the use of IoT is expected to benefit short-range forecasts even before the medium-range.


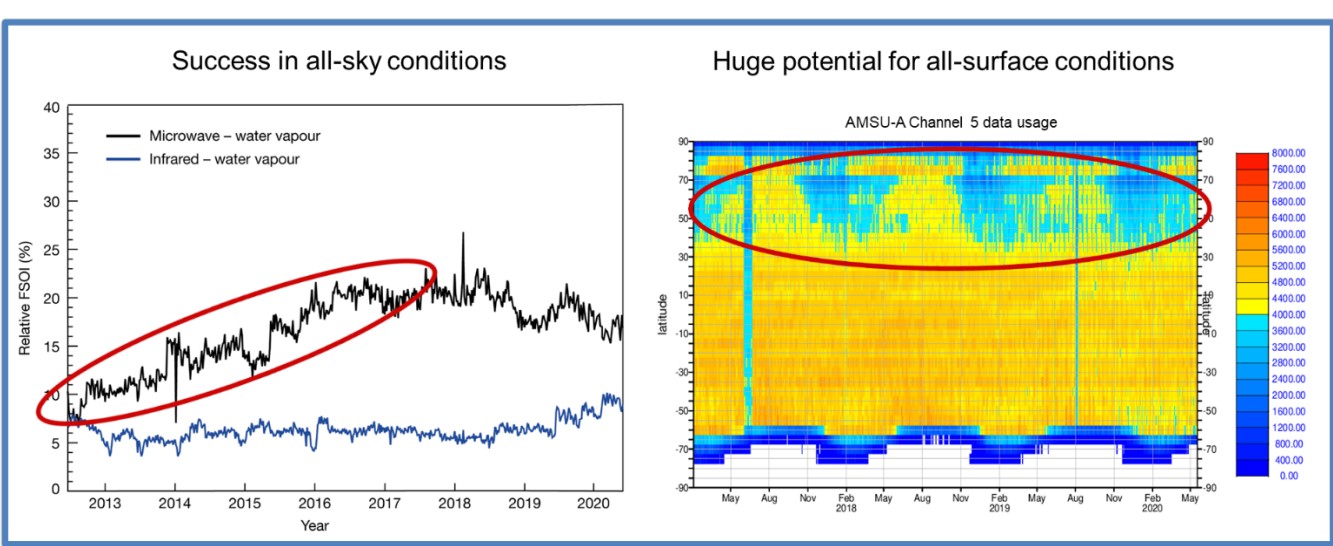

**Figure 1: Better exploitation of satellite observations. The plot on the left shows the relative contribution of microwave water vapour channels to the forecast quality measured with a Forecast Sensitivity to Observation Impact (FSOI) technique. The contribution increased dramatically from little more than 5% to approaching 20%, thanks to the development of solutions to use**
**microwave data in all-sky conditions. The ECMWF's strategy aims to expand this approach to cover all-surface conditions, a challenging but potentially very rewarding research area. The plot on the right shows the microwave radiometer AMSU-A lower tropospheric temperature (channel 5) assimilated daily at different latitudes. In the northern hemisphere, ECMWF manages to**

**assimilate far less data at high latitudes in winter than summer, due to the current limited ability to use the data over surfaces like snow and ice.**


These innovations must be paired with a consistent modelling approach which includes realistic representations of the water, energy, and carbon cycles. This is currently not the case for the water cycle, for example, as in current data assimilation/forecast the mass in the system does not remain constant (principle of mass conservation, (Zsótér et al 2020). The efficient coupling of many of these different processes poses a significant research challenge, closely linked to problems

such as the initialisation of a complex Earth system consisting of components like land, snow, rivers, oceans, and atmospheric compositions. For instance, the provision of the initial conditions of the snowpack on a global scale, which is important for flood forecasts, can only be addressed with more real-time adequate measurements as well as a better physical representation of the processes in the snowpack (ECMWF currently utilizes a single layer snow model).

Another essential aspect of weather forecasting is to capture and represent uncertainties by using ensemble predictions, but

finite computing and requirements to produce timely forecasts only allow a limited number of ensemble members. A chaotic, flow-dependent system will always have to rely on representing uncertainties within some sort of Monte Carlo type framework. More research is needed to better represent uncertainties and, for example, to define the optimal  ensembles size to adequately represent tails of the climate and forecast distribution. Other methods such as post-processing or Artificial Intelligence (AI) will be able to represent such uncertainties to some degree and a careful balance between various methods

has to be found.

Such scientific advances must be supported by substantial technological innovations. Novel HPC architectures for computing and storage are required to improve the representation of processes and uncertainties. At the same time, existing software code and algorithms need to be adapted to take advantage of these new resources and optimize speedup and scaleup of parallel processing. This requires significant investment in computational science, for example in the use of high-

performance, heterogeneous GPU/CPU architectures through domain specific languages, which will allow domain experts (such as natural hazard scientists) to develop programs independently from the underlying architecture (Mernik et al, 2005). Novel architectures (i.e. GPU/FGPU etc) are particularly suited to be used by AI and Machine Learning (ML), which will play an increasingly revolutionary role throughout the entire research and operational chain, from the quality control of observations and approximation of physical equations to speed up execution, to the improved representation of uncertainties

in extreme natural hazards through post-processing (Duben et al., 2021). This is another area where collaboration with the Member States will be fruitful. Finally, we will investigate  novel ways to address the need of increasingly compute-intensive  forecasts and simulations, including computing solutions beyond supercomputers. For example, the OpenIFS@home project allows scientists to exploit spare compute cycles on desktop computers provided by volunteers to run huge ensembles offering researchers a new tool to study weather forecast related questions (Sparrow et al., 2020).

The data produced by these sophisticated systems can however be of use only if they are easily accessible and provided reliably. Future Earth system models and forecasts will generate such large volumes of data that the only feasible way to

extract relevant information will be to bring computing to the data rather than transfer the data to a computing infrastructure (Pappenberger and Palkovic, 2020). This also requires research into novel ways for storing and manipulating big data (Hanley et al., 2020). A typical workflow for a natural hazard forecast or model contains the following components: pre-processing, numerical simulation, post-processing and visualization. Numerical simulations are often run on a High-Performance Computer, whilst the other tasks are more suited to be executed in a cloud environment, although these different technologies will converge in the future. Moving the computation to the data will have the additional benefit of providing an environment that allows systems and workflow components to be co-designed and shared with and by the Member States, stimulating the creation of an eco-system of meteorological data and applications. Research into the efficient management and orchestration of workflows spanning these different computing environments will crucially improve the overall performance of a forecast and model chain. This will allow a larger proportion of limited human and computing resources to be used to improve natural hazard forecasts and models.

## 3 Impact

Advances in science and technology are only relevant within the context of an operational framework if users can maximise the usefulness and accessibility of high-quality products and outputs. Communication and decision-making using big data representing uncertainties remains a significant challenge which will need to be addressed to increase the impact of natural hazard predictions and reanalysis (Neumann et al., 2018, Thielen et al., 2020). ECMWF will aim to provide detailed Earth system simulations of the past, present and future. Although they may be used for different purposes, simulations at different time scales need to be considered in a holistic way. For instance, re-analysis data may be used to establish return periods, which in turn are used to detect weather time anomalies in forecasting or risk-based decisions on climate projections or they may be used simply on their own for a risk analysis. A particular attention will be devoted to extreme events several weeks ahead, to provide skilful predictions of extreme temperature anomalies and hydrological impacts such as droughts and floods, air quality, fires, heat and cold waves, as well as outputs which monitor the environment (i.e., anthropogenic $CO_2$ emissions). Forecasts of natural hazards beyond two weeks require an improved understanding of the sources of predictability and teleconnections (Mastrantonas et al. 2020). In this context, it is necessary to focus strongly on a user-oriented evaluation over multiple temporal scales, variables and coupled components which will inform the optimal pathway of model development. Indeed, user feedback especially from the Member States needs to be more closely integrated into the evaluation chain to allow for cost benefit assessments and ensure that the proportion of resources employed on all elements of the system provides maximum benefits. Policy and decision makers, emergency responders in the Member States and beyond and the public in general will be able to significantly improve their decision making when trust in forecasts is increased through metrics which are understandable and meaningful to them (WMO, 2021, Ebert et al., 2018). A key research challenge is to generate detailed and high-resolution Earth system scenarios able to simulate the effects of different

environmental policies and sustainable development plans to derive optimal conclusions (European Commission, 2021). Each component of the Earth system as well as other factors such as environmental policies have uncertainties attached to them which will interact in complex non-linear ways. Therefore, it is important to fully propagate and cascade all dominant uncertainties through the entire value chain, which is key in supporting efficient user decision making.

## 4 People

Expert and highly motivated people are the most important resource for research or operational activities to improve natural hazard forecasts and models. The working environments are rapidly changing under the pressure and mutual influences of technological advances, globalization and, in the last year, the effects of lockdowns. Successful organisations that aim to create step changes to advance the science and practise in natural hazards need to adapt and harness the opportunities that more dynamic working conditions bring. The expansion of ECMWF's activities in the area of computational science and AI for numerical weather prediction means also that the organisation needs to position itself and become attractive in a highly competitive and rapidly changing job market, which is key to be able to address the challenges mentioned above. ECMWF is becoming a multi-site organisation and as such will be more attractive to potential employees and existing staff as it allows for more flexibility in terms of work places, while aiming to maintain a "One ECMWF" culture. In parallel, ECMWF will continue to modernise its social policies with a particular attention to fostering an appropriate work-life balance, e.g. via flexible working patterns. ECMWF is already an organisation with a strong environmental consciousness as it operates the Copernicus Climate Change Service, which makes it an attractive employer. It is therefore essential to continue enhancing environmental awareness, reducing the carbon footprint and embed environmental awareness even deeper in the organisational culture.

Importantly, all this can be achieved by ECMWF only via a collaborative approach with the European Meteorological Infrastructure as well as other international and national partners, and the private sector.

## 5 Conclusion

ECMWF has embarked on a new strategy aiming to produce cutting-edge science and world-leading weather predictions and monitoring of the Earth system in close collaboration with the members of the European Meteorological Infrastructure, for a safe and prosperous society. An essential part of this strategy is to deliver forecasts of high-impact weather events and natural hazards well into the second week and further ahead. For example, our target is to provide skilful predictions of extreme temperature anomalies and hydrological impacts such as droughts up to three weeks ahead on average. We will need to develop a high-precision digital model of our planet that will make it possible to interactively explore various natural processes and human activities. Advancements in machine learning need to be intertwined with physical based models so that they become indistinguishable from each other revolutionizing the European capability to monitor and predict our

changing planet, based on the integration of extreme-scale computing and the real-time exploitation of all available
environmental data.

In addition, ECMWF aims to support the natural hazard domain through global reanalyses and re-forecasts of weather and environmental hazards to monitor changing patterns and predictability of high-impact events. For example, we will develop anthropogenic $CO_2$ emission estimation systems at global, regional and local scales with a full representation of uncertainties to support the implementation of the international agreements such as the Paris agreement. .

The achievement of this vision will require the adoption of new technologies, the integration of different areas of research, extensive collaboration with National Hydro-meteorological Services and other environmental organisations in Europe and worldwide.

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
