# Peer review of "Invited perspectives: The ECMWF strategy 2021-2030 Challenges in the area of natural hazards"

_Natural Hazards and Earth System Sciences, 2021_

## Author Comment (AC4)

**Response to Reviewer 1**

We thank reviewer 1 for the excellent comments and suggestions, which are very constructive below is a point by point response.

Dear authors,

first of all thank you for your paper, I like it. Congratulations for the vision of bringing code to data, instead of data to code and for open the possibility for AI. Here are my notes, hopefully helpful ones:

corrections:

- Paragraph 35, line 4"form cars" > from cars;

Changed

- Paragraph 45, line 1 "significant research challenge" > a significant research challenge;

Changed

- Paragraph 45, line 1 to 2 "This is closely linked to the challenge of how" > This is closely linked to questions such how to";

Changed

- Paragraph 45, line 3 to 4 "Initialising the snowpack on a global scale which is important for flood forecasts is an example of such a challenge." > maybe just "initialising the snowpack on a global scale which is important for flood forecasts"

Agreed and changed

suggetion:

- Paragraph 40, line 2 "does not conserve mass" >  the mass in the system does not remain constant (principle of mass conservation);

Excellent suggestion for improved readability - done

- Paragraph 45, line 3 to 4 "Initialising the snowpack on a global scale which is important for flood forecasts is an example of such a challenge." > maybe snow calculation module?

  Not sure that would be the right term – will be reworded to "Providing the initial conditions of the snowpack " which hopefully addresses the reviewers comment

- Paragraph 50, line 3 to 4 "which will allow domain experts (natural hazard scientists)" > maybe just natural hazard scientists

Included a 'such as ' to qualify as not all domain experts are natural hazard scientists. Thank you for pointing this out

- Paragraph 50, lines 4 to 5 "GPU based architectures are particularly suited to be used by artificial intelligence and machine learning" > well also FGPAs and ARM-based systems this will require you to repharse the subsequent statements about GPUs...

Excellent point – rephrased to "GPU basedNovel architectures (i.e. GPU/FGPU etc) are particularly …"

- Paragraph 50, line 3 to 5 and Paragraph 60, line 1 "Other novel ways need to be found to address the need of increasingly compute and storage hungry forecasts and simulations and should therefore also include computing solutions beyond supercomputers" > maybe just forecast that are increasingly demanding in terms of computing and storage... computing solutions beyond supercomputers? OpenIFS@home example not familiar and I did not clearly understand your explanation.

This is a good point, removed storage hungry as it is misleading & expanded the explanation of OpenIFS@HOME sligthly

- Paragraph 70, Line 2 "provide maximum benefits to user" > maybe just provide maximum benefits

  Done

- Conclusion is ok, but I would be more bold and incisive in this section.

  OK we have tried to be more bold and incisive by extending into additional areas such as CO2 monitoring and the future digital twin 😉

- rephrase:

- Paragraph 40, lines 2 to 4 "It is also essential to capture and represent in our predictions as many of tahe model and other uncertainties as necessary: for example, we know that many parts of the Earth system are inadequately observed (Beven et al 2020). However, finite computing and requirements to produce timely forecasts will only allow a limited number of ensemble members to represent these uncertainties." > "model and other uncertainties", maybe just uncertainties? > how does "many parts of the Earth system are inadequately observed", better link to the previous statement about uncertainties > "finite computing and requirements", maybe just requirements (or even resources)? > "ensemble members", maybe just complete set of forecasts;

  Deleted the observation constrain paragraph and addressed the issues regarding ensemble member differently (see comments from reviewer RC1)

- Paragraph 45, line 5 and Paragraph 50, lines 1 to 2  "The compute and storage power of novel HPC architectures is required to improve the representation of processes and 50 uncertainties, however existing large code bases to model

and forecast natural hazards are ill equipped to scale for such novel architectures." > maybe like this:  Novel HPC architectures, both for computing and storage, are required to... at the same time, existing codes and algorithms need to be adpated so that they can take advantage of these new resources (speedup and scaleup of parallel processing);

Thank you adapted the rephrasing above

- Paragraph 80, lines 3 to 5 and Paragraph 85, lines 1 to 2 "Computer, whilst the other tasks are more suited to be executed in a cloud environment, although these different technologies will converge in the future. Research into the efficient managing and orchestrating of workflows spanning these different compute environments will crucially improve the overall performance of a forecast and model change and thus allow a larger proportion of limited human and compute resources to be used to improve natural hazard forecasts and models." > too long, please rephrase

Done – thank you

---

## Author Comment (AC5)

**Response to RC2**

The perspective provides a clear picture of the ECMWF's intentions for the next decade. It describes the intended progress from the present practice, describing several key components for a development of capabilities rather than a single step change. These include, in particular, data availability and use, system understanding, computer resources and programming techniques, and human capital.

My impression is that just a few minor improvements of readability of this perspective are possible, which I mentiion below.

Thank you very much!

Abstract: I am aware that the collaboration with the member states (and further partners) is important, but this aspect is not taken up in the rest of the manuscript. Would part of the strategy, for example, include a development of the collaboration with member states beyond the present practice, for example in terms of computing, use of results, or personnel?

We will follow the reviewers recommendation and include more examples through out the text.

Introduction: As the conclusions mention forecasts of high impact weather at the medium-range time scale of a couple of weeks, I wonder if they it would be worthwhile also to mention them in the introductory section.

Done

Science and Technology: Is the mentioning on remote sensing from satellites and IoT-data, but not of Radar, intentional?

An oversight, thank you for pointing it out

People: While the job market aspect is clearly important, I wonder if cooperation aspects (e.g., with the member states) might be worthwhile to be briefly mentioned.

Excellent point - included

---

## Author Comment (AC7)

The manuscript "Invited perspectives: The ECMWF strategy 2021-2030 Challenges in the area of natural hazards" provides a very interesting concise overview of the ECMWF strategy in respect to natural hazards. The description of the envisaged step changes necessary to tackle the identified key challenges for the future are interesting for the international scientific community also beyond Europe. The challenges are quite generic and relevant not only for ECMWF but for many organisations dealing with forecasting weather and resulting natural hazards, as well as with climate reanalyses and projections. The described step changes can inspire scientific studies and advancements in various domains related to natural hazards.

We thank the reviewer for reading our manuscript and these comments

From my point of view, this manuscript is already valuable and well written, however, with a stronger harmonization of terms, an even more clear structure, particularly a closer link between the identified challenges and the envisaged solutions, this manuscript can become significantly more interesting for the scientific community. Thus, I suggest the following:

Abstract: I would find it more convincing if first the challenges are listed, and then afterwards the goals (which are set to tackle these challenges). When reading the goals, I would expect, that you/ECMWF see the additional challenges of designing user specific products and means of communication and decision support under high uncertainties (or similar).

We have changed the abstract and put the challenges first and added a sentence on products & decision making

I suggest to group the goals into the three pillars already in the abstract, this would prepare better for what is coming later. Additionally, I suggest to use harmonized terms, not all interchangeably "goals", "step changes", "vision".

Removed goals from text, but felt that step change and vision are two separate things

I suggest to provide a closer link between the challenges and the envisaged solutions. E.g. in the part 2 science and technology:

Line 43: "parts of the Earth system are inadequately observed (Beven et al 2020)." -> would be good to mention which ones, so that one has at least an idea without needing to read the paper by Beven.

 Removed line in response to other reviewer

Line 43: "However, finite computing and requirements to produce timely forecasts will only allow a limited number of ensemble members to represent these uncertainties." -> are there other means of representing uncertainties, considering uncertainties besides ensembles? What it's the suggestion or idea how to tackle this challenge?

This is an excellent question, the short answer is that, we will need significant more research and careful research design to address the question. A chaotic, flow depended system will always have to rely on presenting uncertainties within some sort of Monte

Carlo type framework, future challenges do need to include not only research into ways how uncertainties are presented but also how large such ensembles have to be to adequately represent tails of the climate and forecast distribution. Other methods such as post-processing or AI will be able to represent such uncertainties to some degree and a careful balance between these methods has to be found. Thank you for the clarification – we fully agree that there is an importance in establishing the number of ensemble members or better represent uncertainties. This will be worked into the relevant paragraph of the document.

Line 47: "Initialising the snowpack on a global scale which is important for flood forecasts is an example of such a challenge." -> would be great if first ideas of how to tackle this challenge can be presented.

Done

Part 3 Impacts:

Line 63: "ECMWF will aim to provide detailed Earth system simulations of the past, present and future with a particular focus on extreme events for several weeks ahead" – This sentence is confusing. I guess that you have two separate tasks, one is rather long-term simulations of the earth system in the past and future, e.g. providing climate change projections for the future, also future scenarios of natural hazards, etc. The other one is early warning of natural hazards and for this the aim is to extent the lead time to more than two weeks. In case this is correct, it would be good to write this more clearly in the manuscript, and structure the impacts part accordingly.

Thank you – separated the sentence which should make it more clear

In this respect, I also guess, that you have different user groups for these two activities? It would be good if you could write a sentence about who your user groups are (and maybe for which products).

This is a good point, but nearly impossible. The separation is less on time scales and we have many customers and users who use both intermixed. So we don't really agree in classifying them as two activities. Example: reanalysis is used to establish return periods, which are used for detecting weather time anomalies for forecasting or risk based decision on climate projections or just on their own for a risk analysis. We actually believe that it is a mistake to make such a crisp distinction. However, we will extend the paragraph to make some more clear statements on the type of usage.

The idea of a user-oriented evaluation seems interesting, it would be interesting to know how this could be realized and how it would influence your work/products. Is it meant to be further developed towards co-design of products together with the users?

One has to be careful from separating user centric design and user centric evaluation, which overlap but are not the same. User centric evaluation is about trust and decision making based. It focuses on *the profile of accuracy and value through the forecasting, warning & communication chain with an emphasis on the information required by decision makers to build their trust in the information they receive*" (HiWeather Research Theme, User oriented Evaluation, http://hiweather.net/Lists/59.html) – added context and references

Lines 77-86 seem to belong more to the technology part, or it needs to be more focused on the visualization and communication aspect.

The paragraph will be spilt and rearranged to reflect the excellent reviewers comment

Line 85 "model chain" not "model change"

Changed – thank you

The part 4 on people is limited to the identification of the challenge. Any ideas for the way forward?

Paragraph will be extended highlighting a way forward, which range from training, communication and sense of belonging to one organisation. We will also highlight the important role the wider European Meteorological Infrastructure plays to foster and maintain talent.

---

## Author Response (AR1)

Point by point response has been provided in the discussion